# Expanding Domain Knowledge Elements for Metro Construction Safety Risk Management Using a Co-Occurrence-Based Pathfinding Approach

**Na Xu** [1] , **Bo Zhang** [2,*] , **Tiantian Gu** [1], **Jie Li** [3] **and Li Wang** [1]

1 School of Mechanics and Civil Engineering, China University of Mining and Technology, Xuzhou 221000, China
2 School of Computer Science and Technology, China University of Mining and Technology, Xuzhou 221000, China
3 School of Civil Engineering, Neijiang Normal Universities, Neijiang 641100, China
* Correspondence: 4090@cumt.edu.cn

**Abstract:** Knowledge is a contribution factor leading to more effective and efficient construction safety management. Metro construction practitioners always find it difficult to determine what specialized knowledge is needed in order to lead to better safety risk management. Currently, domain knowledge elements are generally determined by experts, which is coarse-grained and uncomprehensive. Therefore, this paper aims to provide a structure of domain knowledge elements, using an automatic approach to expand domain knowledge elements (DKEs) from a big dataset of unstructured text documents. First, the co-word co-occurrence network (CCN) was used to find the connected knowledge elements, and then the association rule mining (ARM) was compiled to prune the weakly related subnetworks, leaving the strong associated elements. Finally, a list of DKEs in the metro construction safety risk management was obtained. The result shows that the obtained DKEs are more comprehensive and valuable compared to previous studies. The proposed approach provides an automatic way to expand DKEs from a small amount of known knowledge, minimizing the expert bias. This study also contributes to building a fine-grained knowledge structure for metro construction safety risk management. The structure can be used to guide safety training and help knowledge-based safety risk management.

**Keywords:** metro construction project; safety risk management; knowledge expansion; co-occurrence analysis; association rule mining

## 1. Introduction

Metro construction has presented a powerful momentum for rapid economic development worldwide [1]. Due to various uncertainty factors, especially complex underground geological conditions, metro construction is inherently complicated and high-risk. Many safety accidents and near-miss events are related to ineffective risk management [2,3], resulting in serious social impact, many casualties, and huge economic losses [4]. It is therefore significant to improve the safety risk management in metro construction to avoid and reduce safety accidents and near-miss events.

The architecture, engineering, and construction (AEC) industry is becoming increasingly knowledge-driven and information-intensive [5], especially for metro construction safety risk management, owing to its characteristics of complexity. Research and practice (such as by health and safety executives in the UK) have indicated that up to 80% of accidents are attributed to the actions or omissions of people [6,7]. One of the main causes behind unsafe behaviors is a lack of domain knowledge [8,9], i.e., specific subject-matter knowledge [10]. Hence, increasing domain knowledge leads to the promotion of the level of safety management and a decrease in the rate of accidents and injuries.

Owing to the one-off nature of metro construction projects and organizations, knowledge learning is required every day to meet the demands of addressing new issues. The term domain knowledge elements (DKEs) (e.g., soil mixing wall and shield cut) refers to knowledge that has complete logic and cannot be divided [11]. DKEs can be used as the domain-particular concepts (i.e., knowledge structure) in a domain knowledge graph (DKG), which underpins the knowledge base of safety risk management [12]. However, study on DKEs has received little attention in the literature and practice. Metro construction practitioners always find it difficult to determine what specialized knowledge is needed to lead to better performance and fewer errors.

Since domain knowledge is immense and new DKEs are generated every now and then, the huge mass of information has resulted in a form of information overload for safety risk managers. It is easy to list some DKEs, but it is hard to list all DKEs and update the list in time. Information in text format remains a greatly underutilized form of knowledge in the metro construction safety risk management domain. Hundreds of research works are uploaded to public websites every day, containing rich but unstructured domain knowledge elements. Consequently, this situation calls for an automatic approach for expanding immense unknown DKEs from a small set of existing DKEs.

The main contributions of this work are summarized as follows:

1.  Practically, we built a fine-grained knowledge structure for metro construction safety risk management. The structure can be used to guide safety training and help the construction of domain knowledge graphs, etc.
2.  Theoretically, we propose an automatic approach to expand domain knowledge elements from massive documents, minimizing the expert bias.

In the following sections, a literature review is first given on related research. Then, a pathfinding approach of knowledge expansion is proposed, followed by a step-by-step experiment and its results. Lastly, conclusions are drawn, informing the reader of opportunities for future research.

## 2. Literature Review

### 2.1. Knowledge-Based Safety Risk Management in Metro Construction

Knowledge management research in the AEC industry has significantly blossomed in the last two decades [13]. Since ineffective risk management in metro construction projects is partly due to a lack of knowledge [14], knowledge-based safety risk management is becoming an important method for risk prevention and mitigation. Substantial progress has been made recently. Studies can be separated into three major categories.

The first category centers around using effective knowledge management to increase organization performance regarding safety risk management, such as using knowledge sharing to improve the safety climate [15], exploring knowledge transfer factors to benefit cooperation networks [16], and using a knowledge dynamics-integrated map to clarify the fluidity of knowledge through the risk management process [17].

The second category focuses on knowledge-based intelligent systems to implement safety risk management processes, including automatic risk identification, supervision, and warning. For instance, Ding et al. developed a safety risk identification system for metro construction from construction drawings [18]. Zhong et al. extracted safety risk factors from construction specifications and developed an ontology-based system to match the potential hazards implied in photography images [19]. Current research mainly extracts specified knowledge units related to risk factors and their attributes, e.g., in Ref. [19] construction equipment and its quality, materials, and bearing were identified and extracted as knowledge units.

The third category explores domain knowledge elements. As the core component of knowledge-based systems, a knowledge base is a warehouse of domain-specific knowledge [20]. Four types of knowledge elements were mentioned that lead to successful projects in the AEC industry: Technical fundamentals, materials of construction, construction-applied resources, and field construction operations [21]. Also, key phrases (i.e., domain-specific

compounds of words) were extracted from unstructured text documents with relations based on association frequencies of co-occurring word pairs [22]. Another interesting study put forward a building information modeling (BIM) body of knowledge (BOK) to present common knowledge, skills, and abilities using the Delphi method [23,24].

It is acknowledged that knowledge-based safety risk management is an important and effective method to assist metro construction safety. Knowledge-based intelligent systems have been developed to address safety risk issues. Yet the establishment of DKEs was mainly based on empirical data collected from experts. It is expected that this study helps facilitate the automatic construction and expansion of DKEs.

### 2.2. Automatic Methods for Safety Knowledge Discovery

Benefiting from the big data and artificial intelligence technologies, many automatic methods have been developed to deal with knowledge discovery. The current study mainly focuses on the two categories: (i) data-driven safety risk identification and analysis, and (ii) knowledge extraction from publication works.

Data-driven safety risk analysis is prone to integrate data mining technologies and risk assessment models. Na et al. improved the term frequency (TF) model with information entropy values to extract safety risk factors from construction accident reports [25]. Zhipeng et al. utilized Cramer's V and Phi coefficients to uncover statistical correlations between risk factors [26]. Alshboul et al. [27] combined machine learning (ML) techniques and multiple linear regression (MLR) to predict liquidated damages for construction projects. Wen-hui et al. proposed a comprehensive risk assessment framework incorporating credal networks (CNs) and an improved evaluation based on the distance from average solution (EDAS) method [28]. Additionally, many hybrid models have been developed to improve the accuracy of risk evaluation. Alshboul et al. complied genetic algorithms to optimize the associated decision variables for earthmoving equipment [29]. Zhang et al. optimized t-squares support vector machines (LSSVM) using quantum-behaved particle swarm optimization (QPSO) to perform early risk warning in subway station construction [30]. Shehadeh et al. developed a Gaussian mixture model to estimate the construction companies' capabilities in performing construction and maintenance activities during the pandemic [31]. Li et al. provided a second-order structural model using structural equation modeling (SEM) to determine the safety level in metro subway projects [32]. However, domain knowledge elements are far broader than safety risk factors. Knowledge units required for safety risk management should also be explored, such as the description of metro structures and construction equipment. Hence, the amount of processed data in this study is much larger.

Information extraction (IE) methods are mainly used to extract knowledge from scientific publications [33,34]. Two prevailing tasks of IE are named entity recognition (NER) and relation extraction (RE). The NER approach focuses on finding and classifying relevant knowledge units at a semantic level [35], such as names, organizations, and locations, whereas RE extracts the relationships between entities [36]. NER tasks require highly accurate and domain-specific part-of-speech (POS) tagging results [37], which is laborious and time-consuming. Thus, the linguistic characteristics of DKEs are more complicated. As for RE, the relationship types are fixed and limited within a range of semantic predefined rules, such as IsA, SubClassOf, and AtLocation [38]. For example, Yoo and Jeong utilized ConceptNet to extract relationships, including RalatedTo, IsA, part of, and HasA, between existing words and neologisms from news sites and social media, in order to add new neologisms to existing knowledge [39]. These studies utilized verbs in sentences to extract specified results based on a semantic labeling system, using text mining and natural language processing methods. If two entities in one sentence are related to the main verb, and the main verb is included in the predefined verb lexicon, those entities and relationships are annotated and extracted [40]. However, only very few sentences in the literature, such as definitions, adopt the narratives with specified verbs. Most of the related knowledge elements appear in one document rather than in one sentence.

Thus, a novel method needs to be brought forward to address the two issues of knowledge extraction in the metro construction domain: (i) the obtained DKEs should cover as many knowledge units as possible, and (ii) the process should use as little manpower as possible. To achieve such demands, this study aimed to provide an automatic approach for expanding immense unknown DKEs from a small set of seed words.

## 3. Methodology

### 3.1. Co-Word Co-Occurrence Analysis

Among the various NER and RE techniques, many studies have been based on co-word co-occurrence analysis [41,42]. In the AEC industry, CCNs are widely used to extract and visualize the potential relationships of topics and keywords from large-scale literature works in order to find research trends and gaps [43,44]. The advantage of CCNs is that they visualize the knowledge element network. However, they are considered to typically have limitations in terms of the quality of keywords and the selection of strong linkage. Regarding keywords, the results from indexing are more akin to the conceptualizations of indexers than to those of the scientists whose work is being studied. As for the linkage strength, the frequency of co-occurrences is counted to evaluate the strength of linkages in co-occurrence relationships. For example, in Ref. [43], the number of articles in which two topics tended to co-occur was calculated to evaluate the interlinkage strengths among all topics. Moreover, social network analysis (SNA) was put forward to calculate the linkage of nodes. The density and centrality of high-frequency words were counted to measure the co-occurrence strength [45,46]. However, neither of the above methods is capable of dealing with voluminous data.

To overcome the limitations of CCNs and to enhance the expanding performance, the proposed co-occurrence-based pathfinding approach made the following improvements:

(1) To enlarge the scope of the dataset, a web crawler was utilized to search the literatures on the World Wide Web across related domain platforms. Moreover, the entire abstract was retrieved and used to build a corpus to mitigate the bias of improper keywords. Moreover, because the collected abstracts were unstructured text, a domain lexicon closely related to domain-specific documents was constructed to improve the performance of text segmentation.

(2) A CCN was generated by a huge binary matrix. Both co-occurrence frequency and the centrality of the node have limitations in dealing with big data. To accomplish this, ARM, as a typical data mining method, was integrated to evaluate the strength of network linkage and to prune the redundant subnetworks.

### 3.2. Association Rule Mining

Association rule mining (ARM) is a data-mining method that is widely used in commercial settings (e.g., the purchase tendencies of customers) to find interesting associations that often occur in large datasets [47]. In the construction safety risk management field, Ayhan et al. identified the correlations between the attributes and accident types in occupational accidents [48]. Guo et al. analyzed the relationships among unsafe behaviors, worker types, and construction phases [49]. Zhou et al. investigated the associations between safety risk monitoring types and the coupling of risks [50]. In conclusion, ARM is normally used to analyze the coupling safety factors or the co-occurrences of causes and accidents. This paper aimed to use ARM to evaluate the association strength and highlight the strongly linked DKEs in a CCN.

### 3.3. Architecture of the Co-Occurrence-Based Pathfinding Approach

The idea of co-occurrence-based pathfinding is to assume that DKEs frequently co-occur in one document. The method is based on the integration of a CCN and ARM for finding frequently co-occurring pairwise domain-specific terms in a big dataset of text documents. Figure 1 presents the architecture of the co-occurrence-based pathfinding approach.

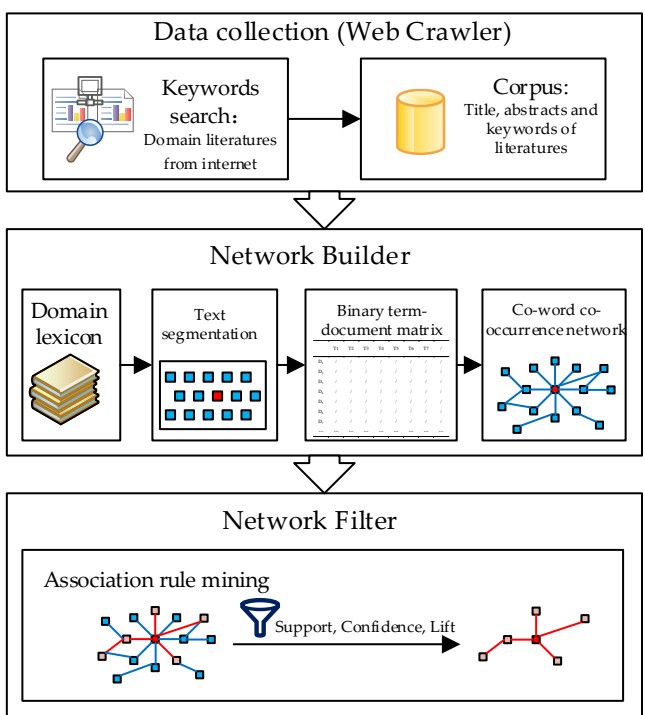

**Figure 1.** Architecture of the co-occurrence-based pathfinding approach.

(1) Web crawler. Select some known DKEs as seed words, and then use the web crawler to collect domain literature, taking the selected seed words as search words. The title, abstract, and keywords of each literature work are collected and stored in the corpus.

(2) Network builder. A general lexicon rarely contains proper or highly technical terms. Thus, a domain lexicon is built to improve the performance of text segmentation. Then, text segmentation is performed and word pairs and a binary term–document matrix (B-TDM) are generated. A CCN is then built based on the B-TDM by counting frequencies of pairwise term co-occurrence.

(3) Network filter. The association rule mining method serves as a filter to remove weakly related subnets in the CCN. Support, confidence, and lift are the three important indicators used for filtering rules by setting thresholds [51]. The threshold values are context-specific and user-defined [52]. The newly discovered DKEs are compared and added to the existing domain knowledge elements by using string matching.

*3.4. Integration of a CCN and ARM*

DKEs usually co-occur in one domain document simultaneously because they describe one subject-specific topic. For example, if the term "geological conditions" appears, then related terms such as "geotechnical structure," "adverse geology," and "collapse accident" may be found in the same document. Therefore, we assumed that word pairs generated by co-occurrence are prone to representing the same specific subject (i.e., domain knowledge). As DKEs and their co-occurrence relationships form a CCN [53], the expansion of DKEs can be achieved by finding the propagated paths of co-occurrence relationships.

Figure 2 displays a CCN of DKEs. The CCN comprises nodes, representing DKEs, and links, representing the co-occurrence relationships between said DKEs. As an example, suppose we want to find unknown DKEs related to the known element A. We begin by exploring the paths from the node labeled A, which leads us to the new nodes B, C, D, E, and F according to the strength of the link. We then expand from node F, leading to node G and subsequently node F. This finding process is repeated until an expansion path no longer appears and the network cannot be expanded any more—at which point it is considered that all domain knowledge elements have been found. Finally, we can find

the following five paths: A→B, A→C, A→D, A→E, and A→F→G→H. Therefore, we can expand the domain knowledge elements from A to B, C, D, E, F, G, and H.

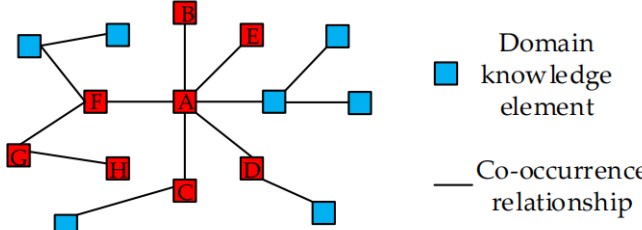

**Figure 2.** Co-word co-occurrence network of domain knowledge elements.

The basic concept of the association rule is to generate rules based on items with frequent occurrence [48]. Association rule mining is utilized here to filter out the weak linage of the CCN based on the support, confidence, and lift value. D is the total number of documents. X and Y represent the two different knowledge elements. An association rule X → Y is defined as: if X occurs, then Y occurs. X is the antecedent node of the CCN, and Y is the node. Support, confidence, and lift are the common indicators employed to evaluate the strength of an association rule [54].

The support shows the number of documents supporting the association rule of co-occurrence (X and Y occur in one document together) over the total number of documents (D). $P(X \cup Y)$ means the possibility of the co-occurrence of both X and Y. As for confidence, it is defined as the frequency with which X and Y occur together over the frequency with which X occurs in isolation. The value of support is an indication of how frequently the word appears in the corpus, while confidence shows how often the co-occurrence is found to be true [55]. Moreover, the value of lift reflects the influence of the occurrence of X on the occurrence of Y. The indicators are defined as in Equations (1)–(3) [51,56].

$$Support_{(X \to Y)} = D_{(X \cup Y)}/D = P(X \cup Y) \tag{1}$$

$$Confidence_{(X \to Y)} = D_{(X \cup Y)}/D_X = P(Y|X) \tag{2}$$

$$Lift_{(X \to Y)} = (D_{(X \cup Y)}/D_X)/(D_y/D) = P(Y|X)/P(Y) \tag{3}$$

The Apriori algorithm is adopted to mine the pairwise item sets X → Y. The path X → Y will remain as a frequent item set when it meets the minimum thresholds of the support, confidence, and lift. Otherwise, the path will be eliminated from the CCN.

Choosing a threshold value is one of the most difficult aspects of applying ARM. Currently, thresholds are determined intuitively by users, according to the dataset's characteristics and the user's desires. Normally, the threshold values of support and confidence are usually set around 1%~4% and 10%~20% in safety risk factor discovery [56], while the threshold value of lift is often set around 1~1.5. In the experiment, statistical analysis was utilized to help generate the candidate threshold groups. Then, the domain experts decided the best threshold group according to the mined outputs. It is expected that the designed process can minimize the user's uncertainty.

## 4. Experiment and Results

### 4.1. Data Collection

Compared to engineering documents in the field, the academic literature is large in terms of quantity of works—more formally, it is easier to obtain and contains more new emerging knowledge. Thus, academic literature work was selected to build the corpus.

(1) Selection of the seed words

Using "metro construction" as the topic word to search the literature in the database of China National Knowledge Infrastructure (CNKI) during the years 2008–2018, we selected the top 100 indexed literature works and extracted the keywords. The, duplicated words

were deleted, synonyms were normalized, and general words were deleted (e.g., control, management, and simulation analysis). Then, a list of seed words (No. = 188) was determined.

(2) Corpus building

With the CNKI, Vepu, and Wanfang databases (three of the main Chinese academic databases) taken as the data sources, web crawling was conducted separately using the seed words as search words to match the keywords in the academic literature published during 2008–2018. Finally, 68,817 literature works were collected, including journal articles, dissertation papers, newspaper reports, and conference proceedings. The title, abstract and keywords of the literature works were collected and transformed to text-type documents to construct a corpus. Moreover, the crawler supports the "Feed Adapter" function for data ingestion, so it continuously integrates data from external sources.

### 4.2. Network Building

According to the architecture (Figure 1) in the Methodology section, four steps were conducted to build a CCN:

(1) Domain lexicon

A domain lexicon was built according to Refs. [57,58]. Not only were subject-specific terms listed in the lexicon, but so were the synonym terms and stopwords. The domain lexicon benefits the generation of subject-specific tokens.

(2) Text segmentation

The corpus was divided into linguistically meaningful units (tokens) in this step. JiebaR, a Chinese tokenization toolkit, was used to implement the segmentation. The created domain lexicon was deployed in the program to improve the performance of text segmentation.

(3) Binary term–document matrix (B-TDM)

B-TDM is a numeric two-dimensional matrix representing the occurrence of a term appearing in a document [59]. JiebaR was used to calculate the occurrence count of each token for each document. The row refers to the sequence number of the document, while the column shows the tokens obtained after text segmentation. The number "1" represents the token occurring in the document, while the number "0" represents the token not occurring. It should be noted that the B-TDM of this case is a large sparse matrix because the distribution of terms is scattered in the massive dataset. Table 1 shows the B-TDM of the case. "Tunnel engineering" and "construction technology" both appear in documents No. 1, No. 2, and No. 5. Therefore, "construction technology" is considered a new candidate DKE related to "tunnel engineering".

**Table 1.** Binary term–document matrix.

|  | Tunnel Engineering | Construction Technology | Shallow Burying | Mining Method | Shield Method | ... | Foundation Support |
|---|---|---|---|---|---|---|---|
| 1 | 1 | 1 | 0 | 1 | 0 |  | 1 |
| 2 | 1 | 1 | 0 | 0 | 0 |  | 0 |
| 3 | 1 | 0 | 0 | 0 | 1 |  | 1 |
| 4 | 1 | 0 | 1 | 0 | 0 |  | 0 |
| 5 | 1 | 1 | 1 | 0 | 0 |  | 0 |
| ... |  |  | ... |  |  |  |  |
| 68,817 | 0 | 0 | 1 | 0 | 1 |  | 1 |

(4) Co-word co-occurrence network

A CCN was created based on the data of B-TDM by counting the frequencies of pairwise term co-occurrence. Because of the large number of redundant co-relations within the tokens, the network needs to be largely pruned to highlight the most important co-related item sets.

### 4.3. Network Filtering

The Arules toolkit was used to perform association rule mining. The B-TDM was taken as the input data, and the Apriori algorithm was chosen to generate the pairwise item sets according to Equations (1)–(3). To set the thresholds of frequent item sets, 10 of the 188 seed words were randomly selected as the training data. Figure 3 displays the support value of 10 seed words by different colors. The mining results reflect that the support value shows a long tail distribution, i.e., only a small number of item sets have a high frequency of co-occurrence. The curve decreases sharply at the beginning, then gradually decreases to a low level after entering the inflection area, and finally forms a nearly horizontal straight line.

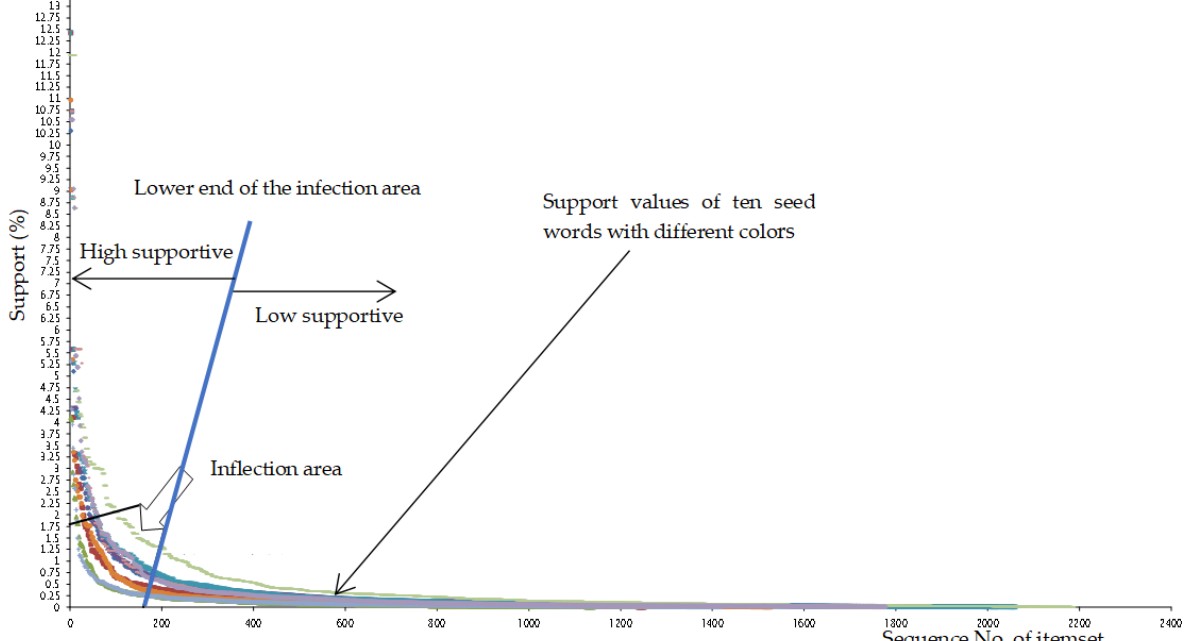

**Figure 3.** Distribution of support values.

According to the dataset's distribution in Figure 3, we attempted to use the inflection area to set around the high/low frequent items. In order not to lose the valuable item sets, the lower end of the inflection area was selected as the boundary to define high and low supportive item sets.

Different seed words have different inflection areas. Figure 4 displays all of the support values at the lower end of the inflection area. To simplify the mining rule, the benchmark of support value was defined as the average of them, which was 0.5%. Thus, the potential mining rule set was defined as the combination of support $\geq \{0.5\%, 1\%\}$ and confidence $\geq \{0, 10\%\}$ and lift $\geq \{1, 1.5\}$. This means that there are eight candidate mining rules. Eight experiments were conducted on the above 10 words, as shown in Figure 3. Expert knowledge was required to evaluate the strong association item sets under different mining rules and to choose the best of them. Finally, the threshold value of the three indicators in this study was set as support $\geq 0.5\%$ and lift $\geq 1$. No limitation for confidence (confidence $= 0$) means that as long as Y occurs, the co-occurrence is valid. Compared to previous studies, the relatively lower threshold value of support and confidence led to more co-related terms. Otherwise, some item sets would have been missed because the B-DTM in this case was very sparse. The value of lift (lift $= 1$) generated those terms whose occurrence increased with the occurrence of the seed words.

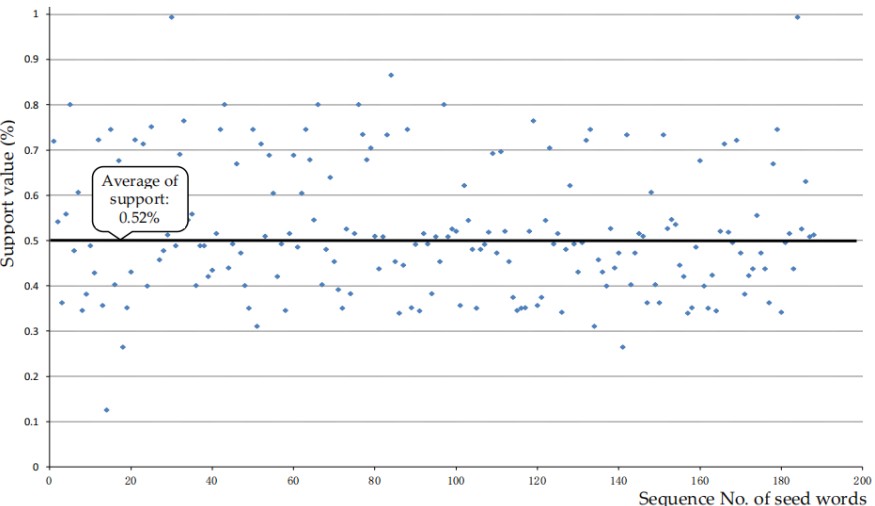

**Figure 4.** Support values of seed words at the lower end of the inflection area.

Therefore, the threshold group was finally set as support $\geq$ 0.5% and confidence $\geq$ 0 and lift $\geq$ 1. The minimum support value is lower than most cases (normally above 1%). This is because of the sparse matrix of B-DTM in this case. It is acknowledged that datasets with a low level of density required a smaller minimum support value when compared to datasets with high density [60].

*4.4. Results*

(1) Knowledge structure of metro construction safety risk management

Applying the association rule to the whole corpus, 2914 strong item sets were obtained. Then, the duplicate terms that already existed in previous rounds of mining were merged by using string matching. Finally, a list of 1583 DKEs was obtained.

To benefit the sharing and reuse of knowledge, the obtained DKEs were grouped into 11 themes and five categories by experts, according to the descriptions and meanings of the knowledge elements. Correspondingly, the knowledge structure of metro construction safety risk management was established (Figure 5). Limited by the length of this paper, the number of DKEs is displayed in parentheses instead of in detail. The knowledge structure can be used in many practical scenarios, such as safety training, ontology establishment in knowledge-based systems, and concept construction in domain knowledge graphs.

(2) The pathfinding process of the proposed approach

To verify the validity of the proposed approach, the process of one of the seed word was taken as an example. Figure 6 displays the co-occurrence-based pathfinding process that the seed word "tunnel engineering" experienced. For the one-round pathfinding, 17 strong associated items related to the word "tunnel engineering" were retrieved. The itemset of bearing capacity and tunnel engineering had the highest support value of 10.11%, indicating that the knowledge of "bearing capacity" is highly related to tunnel engineering in the metro safety risk management domain. Moreover, the knowledge of "construction management" had the highest lift value of 4.32, signifying that it is more likely to appear with "tunnel engineering" than to appear alone. Then, the new collected terms were used as root terms to match other pairwise item sets to retrieve the new DKEs, performing the second, third, and fourth rounds of searching until there were no matching item sets. One item may lead to several new items as long as the itemset meets the predefined mining rules. Finally, 39 new terms were found through the seed word "tunnel engineering".

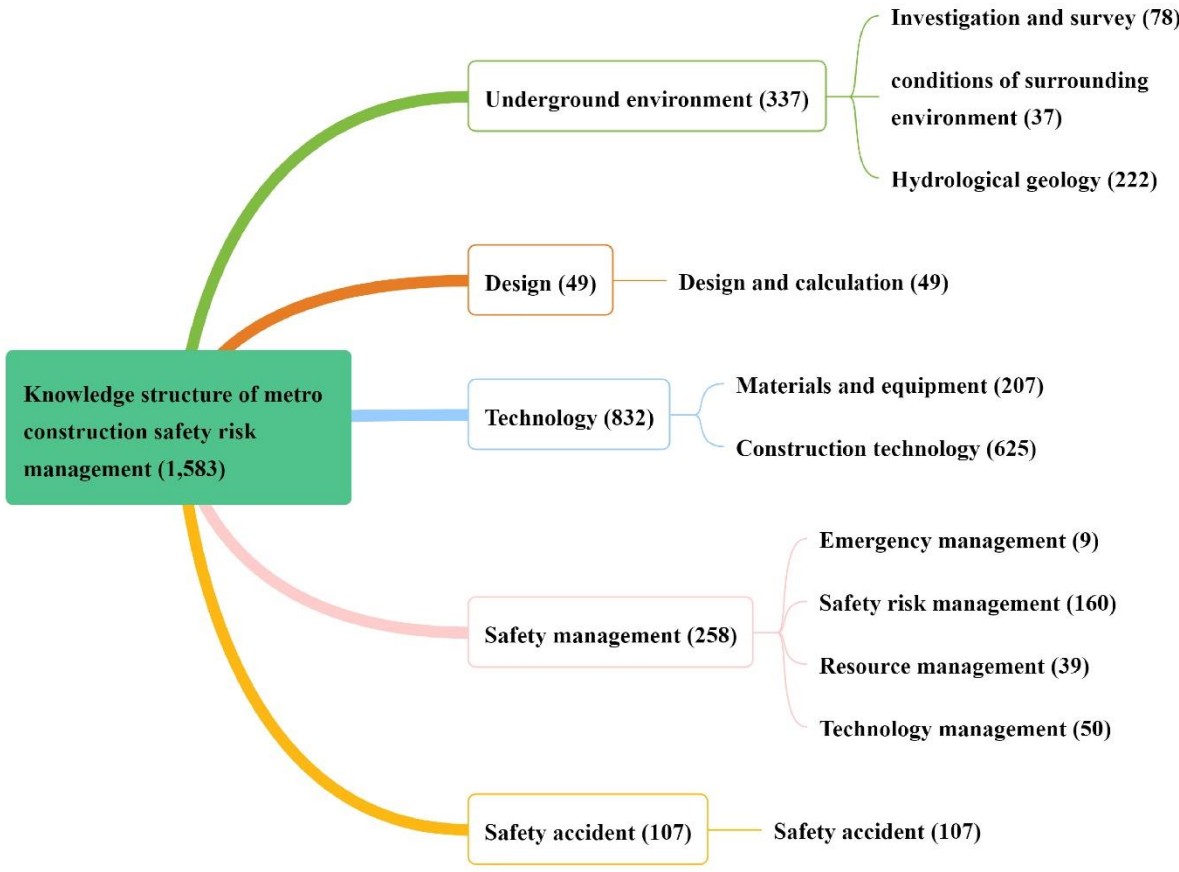

**Figure 5.** Knowledge structure of metro construction safety risk management.

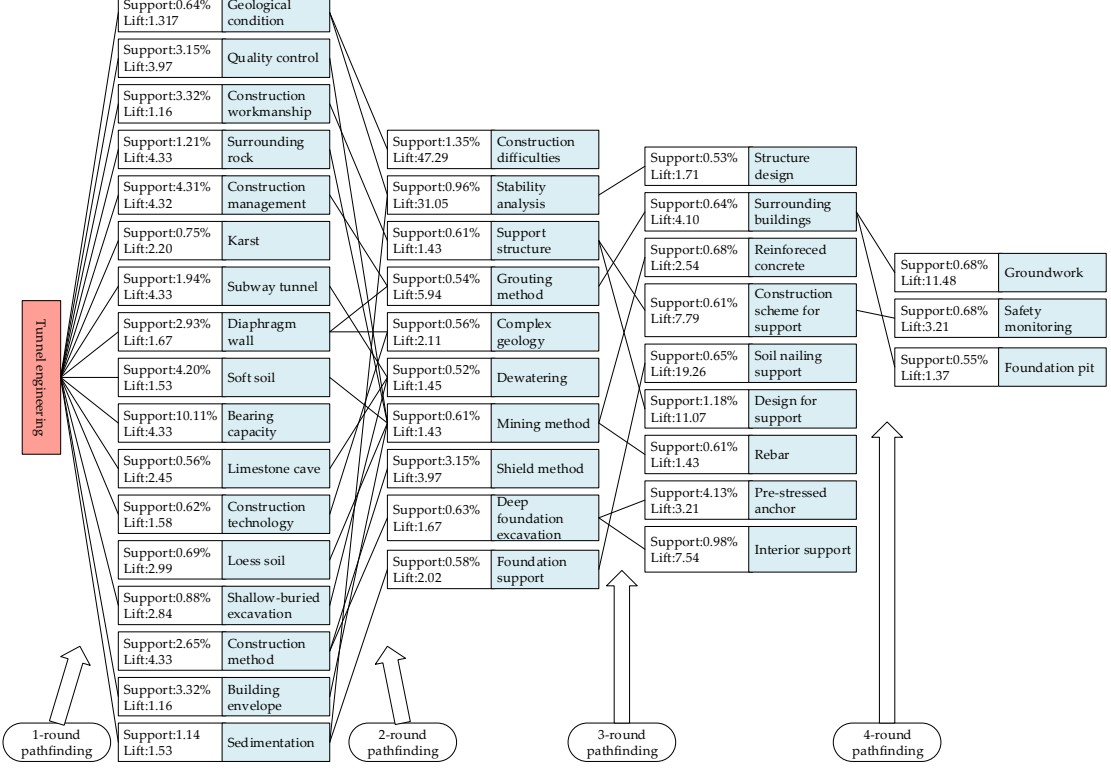

**Figure 6.** Co-occurrence-based pathfinding process for "tunnel engineering".

The result shows that the retrieved knowledge elements are comprehensive and valuable. Additionally, the number of newly found DKEs is rich without being overly redundant, demonstrating that the threshold of the ARM is reasonable. In brief, the proposed pathfinding approach performed well in the experiment.

## 5. Discussion

Knowledge elements are beneficial for the adjustment of the personal knowledge structure and for the acceleration of the process of knowledge sharing and reuse, and even knowledge innovation [61]. It is very important for each organization to reinforce the domain knowledge of technical workers and consolidate the accumulation of knowledge in metro construction. Moreover, extensive knowledge of practitioners can help all participants communicate more easily in order to effectively minimize the sources of risk in metro construction projects [62]. However, knowledge structures were manually established by experts in research and practice. In this study, a systematical and fine-grained knowledge structure is provided for practitioners.

Knowledge on technology takes the lead regarding the number of DKEs, followed by knowledge on the surrounding environment, knowledge on safety management, knowledge on safety accidents, and knowledge on design. Comparisons with previous studies are as follows.

(1) Knowledge on technology

Many previous studies have stated the importance of technical knowledge for construction safety management using questionnaires or expert interviews [63,64]. As Liang et al. stated, technical knowledge and skills constitute the third critical item affecting workers' safety competency, next only to physical conditions and safety awareness [65]. A few studies have tried to increase knowledge on safety behavior and safety conditions to reduce safety risks in metro construction. For example, Guo et al. built a behavioral risk knowledge base for a metro construction project in Wuhan city in order to classify and identify unsafe behavior [66]. Zhou and Ding established a safety barrier warning system for underground construction sites using Internet of Things (IoT) technologies [67].

From the perspective of domain knowledge elements, we confirmed that knowledge on technology, including 625 DKEs about construction technology and 207 DKEs about materials and equipment, occupies the largest number of knowledge elements in the metro construction safety risk management domain. This is probably because technical knowledge can help practitioners discover safety risk factors in the operation context and lead to safe behavior. Similarly, knowledge on materials and equipment, such as tunnel boring machines (TBMs) and shield blades, can help practitioners know better the materials and equipment on construction sites, discover unsafe conditions, and take proactive measures.

(2) Knowledge on the surrounding environment

Many severe safety accidents happen due to the surrounding environment. For example, a tunnel shaft of Xi'an metro collapsed because of excessive excavation and poor geological conditions [67]. Li et al. (2018) determined that underground pipelines are the most frequent reason for metro construction safety accidents based on an analysis of 156 accident reports [59].

We confirmed that knowledge on the underground environment plays a significant role in metro construction safety. This part of knowledge consists of 222 hydrological geology DKEs, 78 investigation survey DKEs, and 37 surrounding environment DKEs. Knowledge on hydrological geology, such as karst, soft soil, and geologic hazards, occupies the second largest number of knowledge themes, indicating that the unique nature of underground hydrological geology is of great uncertainty and the related knowledge plays a very import role for metro construction safety risk management. Knowledge on investigation surveys (advanced surveys, special surveys, and additional surveys, for example) and knowledge on the conditions of the surrounding environment (such as surface settlement and deformation observation) refer to the construction procedures and operation

rules that practitioners should obey. Underground risks are hard to predict and prevent. However, knowledge on the underground environment can help practitioners understand how to perform investigation work and follow the work procedure and standards.

(3) Knowledge on safety management

Safety management is usually considered an indirect reason leading to a safety accident [68]. To decrease such risks, practitioners need to learn and understand how to perform safety risk management, especially the accurate identification of potential safety risks and safety management decision making during the construction process [21].

Knowledge on safety management focuses on knowledge elements about management, including 160 safety risk management DKEs (e.g., risk identification and risk loss), 50 technology management DKEs (e.g., safety inspection and safety supervision), 39 resource management DKEs (e.g., safety training), and 9 emergency management DKEs (e.g., emergency responsibility). This part of knowledge focuses on the management theory, method, and procedure, such as safety inspection and safety risk identification and analysis during metro construction. This part of knowledge may help practitioners build an identify for the organization's safety culture and perform good teamwork.

(4) Knowledge on safety accidents

Learning from past accidents is considered an effective way to prevent the occurrence of similar accidents and to promote construction safety [69,70]. Knowledge on safety accidents, such as accident type, near-miss accidents, and accident causation theory, is needed to identify hazards in the workplace and to take actions to prevent the occurrence of accidents. It is noted that knowledge on proper actions (e.g., knowledge on rescue and recovery) after the occurrence of accidents is also important to prevent further damage.

(5) Knowledge on design

Although design is considered one of the most important potential risks for construction safety [71], knowledge on design seems less important for workplace practitioners compared to other types of knowledge. This may be due to design and construction phrases being separate in most metro construction projects [62]. Little knowledge on design is needed for project managers and workers on construction sites, except for some essential concepts, such as bearing capacity and stability analysis. However, designers need to improve their knowledge of hazards because many construction accidents are connected to the design.

## 6. Conclusions

The current study developed a hybrid model to expand domain knowledge elements (DKEs) from a big dataset of text documents for metro construction safety risk management. First, the CCN was used to build the pathfinding network of candidate DKEs, and then the ARM was compiled to prune the weak related subnets, leaving the valuable ones. A case study was conducted using the Chinese academic literature as the corpus. The result verifies that the proposed approach is applicable to automatically expand domain knowledge elements from a big dataset of text documents. The advantage of the proposed approach is that it minimizes the expert bias.

Moreover, a list of knowledge elements was obtained. Knowledge on construction technology, hydrological geology, and construction resources constitutes the top three largest groups of knowledge elements. They play the most important role in metro construction safety risk management from the perspective of required knowledge. The obtained DKEs compose a fine-grained knowledge structure for practitioners. The knowledge structure can be used in various fields, such as safety education and training, construction of domain knowledge graphs, knowledge-based intelligent systems, and domain lexicon supplementation.

This approach can be extended to other projects to help engineers build a domain knowledge structure. There are three major limitations in this study that could be addressed in future research. First, the quantitative comparison between the method proposed in this paper and other methods is lacking, because the domain knowledge is too immense to

check systematically. Second, the threshold determination of ARM still requires experts' recognition. A more general and automated approach is in need. Third, among the useful item sets extracted from a database, frequent item sets are usually thought to unfold "regularities" in the data [72]. In some situations, however, it may be interesting to search for "rare" item sets. These correspond to new emerging words, which might evolve into very important trends in the future of the domain. Moreover, exploring the semantic relationships of knowledge elements based on natural language processing and deep learning may advance the relation establishment for a knowledge graph.

**Author Contributions:** Conceptualization, N.X. and J.L.; methodology, B.Z.; software, L.W.; validation, T.G. and L.W.; formal analysis, N.X.; investigation, B.Z.; resources, B.Z.; data curation, B.Z.; writing—original draft preparation, B.Z.; writing—review and editing, N.X.; visualization, T.G.; supervision, N.X.; project administration, N.X.; funding acquisition, N.X. All authors have read and agreed to the published version of the manuscript.

**Funding:** This research was supported by the National Natural Science Foundation of China (grant number 71901206) and the Social Science Fund of Jiangsu Province (22GLB023).

**Institutional Review Board Statement:** Not applicable.

**Informed Consent Statement:** Not applicable.

**Data Availability Statement:** Some or all of the data, models, and codes that support the findings of this study are available from the corresponding author upon reasonable request.

**Conflicts of Interest:** The authors declare no conflict of interest.

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
