# Peer review of "Expanding Domain Knowledge Elements for Metro Construction Safety Risk Management Using a Co-Occurrence-Based Pathfinding Approach"

_buildings, doi:10.3390/buildings12101510_

Round 1

Reviewer 1 Report

Thank you for inviting me to review this manuscript. This is an interesting paper that addresses an importation problem in knowledge management. This paper aims to expand domain knowledge elements (DKEs) from big dataset of text documents for metro construction safety risk management. The authors present a novel co-occurrence-based pathfinding approach, which combines the co-word co-occurrence network (CCN) and the association rule mining (ARM). This paper could provide certain insights to people interested in KM in construction. The paper is overall well written. I would recommend publication after addressing some minor comments.

1.       Please indicate the relation between DKE and KG.

2.       Please state clearly your point of departure.

3.       Although not addressed in this paper, please add some statements in the future work about how to enhance the semantic link between DKEs.

4.       Lines 243-246:how do you determine the threshold?

5.       Figure 3 is impossible to read. Please improve the layout of the figure and add more explanation on it.

Reviewer 2 Report

Dear authors, this topic is suitable for publishing, thanks for the opportunity to review. The article has an appropriate structure and comprehensive conclusions.

Author Response

Thanks a lot for the comments. We appreciate it very much.

Reviewer 3 Report

This article emphasizes the importance of expanding domain knowledge elements (DKEs) from a big dataset of text documents for metro construction safety risk management.

1.       First, English requires attention. There are many long sentences, long paragraphs, grammatical errors, and phrases with ambiguous meanings. I strongly recommend the paper is submitted to a professional and reputable English editor for both proofreading and stylistic editing. Otherwise, the meanings conveyed are not clear and the reviewers will not appreciate the novelty and importance of your contributions.

2.      The abstract section requires major improvements in presentation style. For example, the purpose of this paper is completely not clear.  The abstract does not include all findings or contributions this manuscript makes to the state of knowledge and practice. Also, it is recommended to use the presentation style below* to present the abstract.

* A typically good structure includes one or two sentences about:

- What is the problem to be addressed and why it is important?

- Provide a justification for insufficient previous research.

- Tell the reader what you have done.

- Explain "how" you have done it.

- Report what the results are (findings).

- Report what your results mean (contributions)

 3.      In general, the introduction and literature review is a bit on the light side. The amount of work in this area continues to rapidly rise. The authors are advised to strengthen their literature review section with supplementary material and the most updated research in this field; it lacked depth and further reflections. Thus, it is strongly recommended to refer to these papers

https://doi.org/10.1016/j.ssci.2021.105216

https://www.sciencedirect.com/science/article/abs/pii/S0886779822002565

https://doi.org/10.1016/j.asoc.2021.107436

https://doi.org/10.1108/JFM-10-2021-0129

https://www.sciencedirect.com/science/article/abs/pii/S0951832022000588

https://doi.org/10.3390/su14105835

4.      The authors need to clarify their contribution in relation to the available literature

5.      The Methodology section needs more justification and discussion in relation to the selected significant variables.

6.      The limitations of the methodology and results need substantial improvement for the paper to have an impact on research and practice.

7.      The proposed model should be validated against many cases to be validated

8.      The presentation of the results is confusing and not well organized.

9.      It is recommended that you add the practical and theoretical implications section, and how this model would enhance decision-making by providing examples.

10.  The results of your comparative study should be discussed in-depth and with more insightful comments on the behavior of your results in various case studies.

11.  Although the current conclusion description is good, the following key points are still missing:
a.      It is recommended that the conclusions be drawn based on a quantitative comparison of results obtained rather than just mentioning the proposed model. That is, what was compared, how better did it perform, and then leading the reader to explain why it performed better etc.
b.      Critical aspects such as how, where, and why these developed techniques could be useful in real-world construction applications need to be included.
c.      Along with the above, a brief discussion about the broader impacts should also be included. 

Round 2

Reviewer 3 Report

The reviewer would like to thank the authors for the detailed response. However, the following points still unprocessed yet.

1- The literature review should be expanded. I suggest adding the following references:

Li, X., Liao, F., Wang, C., & Alashwal, A. (2022). Managing Safety Hazards in Metro Subway Projects under Complex Environmental Conditions. ASCE-ASME Journal of Risk and Uncertainty in Engineering Systems, Part A: Civil Engineering8(1), 04021079.

Zhang, L., Wang, J., Wu, H., Wu, M., Guo, J., & Wang, S. (2022). Early Warning of the Construction Safety Risk of a Subway Station Based on the LSSVM Optimized by QPSO. Applied Sciences12(11), 5712.

Ali Shehadeh, Odey Alshboul & Ola Hamedat (2022) A Gaussian mixture model evaluation of construction companies’ business acceptance capabilities in performing construction and maintenance activities during COVID-19 pandemic, International Journal of Management Science and Engineering Management, 17:2, 112-122, DOI: 10.1080/17509653.2021.1991851

2- I suggest adding a separate paragraph in the conclusion to aggregate the study’s theoretical and practical implications. 

Best of luck. 
